# Metagenomics-Guided Survey, Isolation, and Characterization of Uranium Resistant Microbiota from the Savannah River Site, USA

**DOI:** 10.3390/genes10050325

**Published:** 2019-04-28

**Authors:** Rajneesh Jaswal, Ashish Pathak, Bobby Edwards III, Robert Lewis III, John C. Seaman, Paul Stothard, Kirill Krivushin, Jochen Blom, Oliver Rupp, Ashvini Chauhan

**Affiliations:** 1School of the Environment, 1515 S. MLK Blvd., Suite 305B, Building FSHSRC, Florida A&M University, Tallahassee, FL 32307, USA; rajneesh.jaswal@famu.edu (R.J.); ashishpathak72@gmail.com (A.P.); edwardsb89@gmail.com (B.E.III); trey.lewis@comcast.net (R.L.III); 2Savannah River Ecology Laboratory, The University of Georgia. PO Drawer E, Aiken, SC 29802, USA; seaman@srel.uga.edu; 3Department of Agricultural, Food and Nutritional Science, University of Alberta, Edmonton, AB T6G2P5, Canada; stothard@ualberta.ca (P.S.); krivushi@ualberta.ca (K.K.); 4Bioinformatics and Systems Biology, Justus-Liebig-University Giessen, 35392 Giessen, Germany; Jochen.Blom@computational.bio.uni-giessen.de (J.B.); Oliver.Rupp@computational.bio.uni-giessen.de (O.R.)

**Keywords:** uranium resistance, metagenomics, whole genome sequence, comparative genomics, *Burkholderia*, Penicillium, diffusion chamber, microbial trap

## Abstract

Despite the recent advancements in culturomics, isolation of the majority of environmental microbiota performing critical ecosystem services, such as bioremediation of contaminants, remains elusive. Towards this end, we conducted a metagenomics-guided comparative assessment of soil microbial diversity and functions present in uraniferous soils relative to those that grew in diffusion chambers (DC) or microbial traps (MT), followed by isolation of uranium (U) resistant microbiota. Shotgun metagenomic analysis performed on the soils used to establish the DC/MT chambers revealed Proteobacterial phyla and *Burkholderia* genus to be the most abundant among bacteria. The chamber-associated growth conditions further increased their abundances relative to the soils. Ascomycota was the most abundant fungal phylum in the chambers relative to the soils, with *Penicillium* as the most dominant genus. Metagenomics-based taxonomic findings completely mirrored the taxonomic composition of the retrieved isolates such that the U-resistant bacteria and fungi mainly belonged to *Burkholderia* and *Penicillium* species, thus confirming that the chambers facilitated proliferation and subsequent isolation of specific microbiota with environmentally relevant functions. Furthermore, shotgun metagenomic analysis also revealed that the gene classes for carbohydrate metabolism, virulence, and respiration predominated with functions related to stress response, membrane transport, and metabolism of aromatic compounds were also identified, albeit at lower levels. Of major note was the successful isolation of a potentially novel *Penicillium* species using the MT approach, as evidenced by whole genome sequence analysis and comparative genomic analysis, thus enhancing our overall understanding on the uranium cycling microbiota within the tested uraniferous soils.

## 1. Introduction

A myriad of ecosystem services are performed by soil microbiota, including biogeochemical cycling of nutrients, degradation of contaminants and as symbionts, and enhancing both animal and plant health [1]. However, despite making tremendous progress in microbiological techniques, microbial ecologists are yet to successfully culture the vast majority of environmental microbiota. In fact, it is suggested that less than 1% of the total microbiota is amenable to cultivation under standard laboratory conditions [2]. A variety of bottlenecks preclude successful isolation of soil microbiota to include, but not limited to, soil dilution prior to plating out, which may eliminate proliferation of those bacteria that exist in extremely low numbers and yet may provide a significant function. Media components that facilitate faster growth of certain bacteria, called copiotrophs, relative to the slow growing oligotrophs and, as shown in some studies, the soil microbiota may require cofactors or cell-to-cell signaling molecules usually produced by bacterial communities in their native habitat, which are not available under standard laboratory growth conditions and hence preclude successful isolation of the vast majority of environmental microbiota [3]. To improve microbial growth and isolation from environmental samples, common approaches, such as varying media composition and growth conditions, have been developed [4]. Others approaches significantly deviate from traditional culturing, such as high throughput extinction culturing [5], high throughput single-cell encapsulation [6], diffusion chambers [7], using different gelling agents, antioxidants, or signaling molecules [8,9], and even increasing incubation times along with reducing nutrient concentrations or simple alterations in the media preparation [10,11,12]. Another interesting technique recently shown to retrieve previously uncultivable microbiota from environmental sample entails soil printing, called biological laser printing (BioLP), which provided a deeper understanding on the nature of microbiota colonizing a miniscule soil niche of 100 μm in size [13].

Furthermore, with the application of 16S ribosomal gene sequencing-based metagenomics or shotgun metagenomic approaches, environmental niches can now be probed to obtain a precise and sensitive understanding of the native microbial diversity along with the plethora of ecosystem services they provide, such as detoxification of environmental contaminants in their native habitats. As examples, metagenomics provided a deeper understanding in soils contaminated with heavy metals and radionuclides [14,15,16]. In fact, previous workers have shown that 90% of bacterial members taxonomically affiliated with Firmicutes, Gammaproteobacteria, Actinobacteria, Bacteroidetes, and Betaproteobacteria resisted up to 4 mM of uranium [17]. Kulkarni et al., isolated two phosphatase producing bacteria *E*. *coli* and *Deinococcus radiodurans*; both isolates grew in the presence of up to 20 mM uranium [18]. Furthermore, it is also known that fungal assemblages possess the metabolic ability to outcompete bacteria under high environmental stresses, including high uranium concentrations, mainly by binding uranyl ions to phosphate and carboxyl groups present in the fungal cellular wall, [19,20] but the information on fungal diversity and molecular mechanisms to circumvent U toxicity is scarce. Towards this end, recent sequencing efforts on *Penicillium* genomes are beginning to provide critical insights into the genomic and metabolic diversity of this soil-borne genus [21,22]. Thus, genomic and metagenomic techniques can be collectively used as sensitive and precise guiding tools to gain valuable insights into the plethora of both, bacterial and fungal assemblage diversity and their metabolic functions, paving the path towards their isolation and downstream ecological and environmental applications.

However, in tandem with the above stated omics based analyses, sensitive and precise techniques are also needed to isolate specific microorganisms, identified by molecular surveys, so that the microbially-mediated functional traits can be better studied and understood for appropriate downstream applications, such as bioremediation. Towards this end, Bollmann et al. [7] conducted a molecular survey of uraniferous soils collected from the U.S. Department of Energy’s Field Research Center (FRC) in Oak Ridge, TN and isolated several potentially novel bacteria with the ability to resist uranium (U) using diffusion chambers (DC). Moreover, the number of DC isolated strains were significantly larger relative to those obtained by direct plating of samples. The fundamental premise of a diffusion chamber is based on the cultivation of environmental microbiota facilitated either in situ [23] or under controlled laboratory conditions in a chamber that simulates the extant environmental growth conditions. Thus, microbial growth is facilitated via nutrients and other molecules seeping into the chambers from the bottom layer of moist native soils, as well as permitting microbial interactions to occur [7,24]. The DC approach has already revealed interesting findings from environments that range from soils, sediments [7,24], and even marine sponges [25].

However, such studies have generally focused on the isolation of bacterial communities, with soil fungi continuing to be largely ignored. Note that a growing body of literature now shows that fungi often outcompete bacteria at high concentrations of environmental contaminants, specifically U [20] and other contaminants as well [26,27,28]. Hence, to take advantage of this physiological attribute, both bacteria and fungi are being aggressively pursued as possible agents for environmental mitigation of uranium and providing better stewardship of historically polluted environments. In fact, a body of information exists on the nature of microorganisms that are not only capable of coping with U stress for their survival but which also detoxify the radionuclide using strategies based on cellular bioreduction, biosorption, biomineralization, and bioaccumulation of uranium [29]. Among these, phosphatase enzyme-based biomineralization has recently garnered significant interest because this bioremediative process has the potential to convert the highly mobile and toxic U(VI) species into a stable and poorly mobile mineral state within the environmental matrices [30].

Among the more well-known molecular mechanisms that underpin bacterial and fungal response(s) to uranium, included are the overexpression of a phytase enzyme and an ABC transporter in *Caulobacter crescentus* [31]. Another study found a suite of 591 proteins that differed significantly in abundance when *Microbacterium oleivorans* A9 was grown in the presence or absence of uranyl nitrate [32]. To further understand environmentally-relevant genomic mechanisms that underpin microbial survival in the Savannah River Site (SRS) co-contaminated ecosystems, we recently isolated several bacterial and fungal strains in the presence of high concentrations of both U and Ni [33,34]. A 16S-gene based analysis revealed that the isolated strains mainly belonged to *Burkholderia* spp. and *Arthrobacter* spp. Both these bacterial genera have been demonstrated to serve as bioindicators of environmental contamination as well as agents of U bioremediation [35,36]. Furthermore, our recent genomic and proteogenomic analyses on several *Burkholderia* spp. and *Arthrobacter* spp. is beginning to unravel the molecular basis for resistance against uranium, including a suite of substrate binding proteins, permeases, transport proteins/regulators, efflux pumps, metal resistance proteins (e.g., for cadmium, cobalt, and zinc), stress proteins, cytochromes, and drug resistance gene determinants, likely working in concert to potentially detoxify uranium toxicity within the SRS impacted ecosystem [33,34,37]. Therefore, it appears that both *Burkholderia* spp. and *Arthrobacter* spp. have a strong metabolic basis for colonization and survival in radionuclide and metal contaminated habitats.

Other than the above stated works on the responses of axenic microorganisms to heavy metals and uranium, only a handful studies are available on the impacts of U stress at the metagenomic levels, especially the shotgun metagenomic response(s) in context with community-wide functional potential [15,38,39,40]. Despite the paucity of information, the functional metagenomic studies conducted, as of this writing, have revealed carbohydrate metabolism and membrane transport functions to be overrepresented in U contaminated soils [40], indicating U likely enhances the microbial utilization of carbohydrates, which may also drive up the respiration rates in the uranium contaminated soils.

Our study site, the SRS, is located near Aiken, South Carolina, where large quantities of U were released to the environment from decades of nuclear materials production and refinement for nuclear weapons production activities [41]. Successful isolation and subsequent studies on U-resistant microbiota can provide a deeper understanding of the cellular mechanisms underpinning bioremediation and metal-microbe interactions that permit U immobilization in the environment. Diffusion chamber-based studies on microbial diversity have previously been used to successfully isolate U-resistant microbiota from the subsurface soils of another DOE contaminated location, the FRC site in Oak Ridge, TN [7]. The use of MT has also been shown to yield a diverse assemblage of soil actinomycetes in comparison to the conventional isolation approach, such that several strains of *Streptomyces*, *Streptacidiphilus*, *Catellatospora*, *Lentzea*, and *Catenulispora* were amenable to isolation [42]. Therefore, in the current study, we developed a slightly modified version of the diffusion chamber (DC) [7] and microbial trap (MT) techniques [43], using uraniferous soils collected from the Savannah River Site (SRS). Such metagenomics-guided assessment and isolation of U-resistant bacterial and fungal strains were undertaken in this study.

The efficacy of the DC/MT technique was successfully demonstrated for (1) identification of the predominant bacterial and fungal communities present in the tested soils and (2) isolation of the predominant bacterial and fungal strains, including a potentially novel species of Penicillia. Overall, this provides a proof-of-concept for the successful application of both the DC and MT techniques to garner a better understanding of structural-functional relationships of environmental microbiota in uraniferous soils. To further gain a detailed understanding on the nature of the newly isolated *Penicillium* species MT2, a draft genome sequence was obtained, which led to a detailed comparative genomics study, thus providing a unique peek into strain MT2′s arsenal of metal-resistance traits. Overall, such DC/MT based studies will be useful for the isolation of native, and potentially novel, soil microbiota that have developed resistance against uranium and, potentially, other environmental contaminants, so that a holistic understanding on metal-microbe interactions and mechanisms that underpin microbially-mediated U detoxification processes are obtained for better stewardship of historically contaminated systems, such as the SRS.

## 2. Materials and Methods

### 2.1. Soil Sample Collection

Triplicate surficial soil samples were collected from an abandoned farm pond, the Steeds Pond, that served as a natural settling basin along the Tims Branch stream corridor on the SRS, located near Aiken, South Carolina. A total U concentration of 4.2 mM was observed in the soil samples used in this study for the establishment of DC/MT chambers [33], which is several orders in magnitude higher than that recognized by EPA to cause toxicity effects [42]. Samples were stored on ice, shipped overnight to the FAMU laboratory, and immediately processed for establishment of diffusion chambers and microbial traps, as well as DNA isolation for metagenomics. All experiments were initiated within a day of sample collection, thus minimizing any artifacts resulting from delayed sample processing for downstream experiments. Chemicals used in this study were the highest analytical grade and purchased from VWR (Atlanta, GA, USA), unless otherwise mentioned.

### 2.2. Establishment of Diffusion Chambers/ Microbial Traps and Isolation of U-Resistant Microbiota

To establish the diffusion chamber, a 1 g soil sample (a mixture of the 3 replicated samples) was mixed with 9 ml of sterile normal saline and serially diluted to 10^−3^ dilution. One ml of this diluted sample was mixed with 9 ml of sterile molten agar at 45 °C to give a final concentration of 10^−4^. This was followed by fixing of a 0.03 µm pore size polycarbonate membrane (GE Healthcare Biosciences, Pittsburgh, USA) glued to the tray (using silicone glue) in such a way that 9 holes on the tray would fall exactly towards the center of the membrane. Instead of working on a single DC/MT chamber, we used a plate with several even sized holes 0.8 mm in height and 0.7 mm in diameter; 9 of such holes were used as multiple chambers, representing DC or MT, to increase throughput microbial cultivation. Approximately 435 µl of the 10^−4^ diluted soil sample in agar was added to each well to fill it completely. After the agar solidified, a second membrane (0.03 µm pore size for DC and 0.2 µm pore size for MT) was glued on top to seal the chamber. Note that the MT approach is used for the isolation of uncultivable actinomycetes, many of which are well-known for their biodegradative and metabolic potential. The premise of this approach is that the 0.03 µm pore-size membrane permits the diffusion of nutrients into the chamber and facilitates the growth of environmental microbiota within the agar by mimicking in situ growth conditions. The DC (any side) and MT (0.2 µm pore size, side facing down) were then placed on the same U contaminated soil (approximately 5 mm thick) and incubated at 28 °C for 20 days. The DC plates were flipped every 2 days and the soil underneath was mixed to remove the buildup of anaerobic conditions. The MT plates were not flipped but the soil underneath was mixed every two days to prevent the buildup of anaerobic conditions. After 20 days of incubation, chambers were opened and the gel-embedded biomass inside the DC and MT were collected and homogenized by passaging through a 22 gauge needle syringe, twice. This was called generation 1 (Gen1), part of which was used for DNA isolation as well as inoculation of the second-generation chambers (called Gen2), which were further incubated for 20 days. Bacteria and fungi from the Gen1 and Gen2 chambers were isolated on Luria Broth (LB) media supplemented with uranium, as well as soil extract media, to retrieve those isolates that are not able to grow with the high nutrients contained in the LB media. The soil extract media was prepared by modifying the method used by Bollman et al. [7]. Briefly, soil sediments were mixed in DI water at 100 g/L concentration and mixed on a magnetic stirrer for 1 h. The resulting solution was filtered through a Whatman filter paper (No. 54) to remove all the particulate matter. The resulting solution was solidified by the addition of 1.5% Agar (VWR, Atlanta, GA, USA) to prepare soil extract media plates.

### 2.3. Microbiome Analysis

Genomic DNA was isolated from soils and DC/MT plugs using the DNeasy PowerLyzer Kit, according to the manufacturer’s instructions (Qiagen Inc., Germantown, MD, USA). Quantity and quality of the isolated genomic DNA was evaluated by using a micro-volume spectrophotometer (NanoDrop Technologies, Wilmington, DE, USA) and further processed for shotgun metagenomics. Briefly, sequence libraries were prepared using the Illumina Nextera XT kit, according to manufacturer’s instructions (Illumina Inc., San Diego, CA). Sequencing was performed on an Illumina NextSeq500 instrument employing a mid-output kit with 2 × 150 paired-end sequencing. For taxonomic profiling, raw reads were mapped to the NCBI non-redundant protein database using DIAMOND [44]. Taxonomic summaries per read were obtained using MEGAN’s Least Common Ancestor algorithm [45] and then summarized across all reads to create counts per taxon. Functional profiling was performed using SUPER-FOCUS [46], with parameters “-a diamond -db DB_100 -n 0”. In both cases, raw counts were normalized to counts per million (CPM) units for relative abundance estimates. Stacked bar-plots were generated on the relative abundance estimates from each sample for taxonomic estimates for phylum and genus separately for bacteria, archaea, fungi and functional subsystem levels 1, 2, and 3, respectively.

MicrobiomeAnalyst pipeline [47] was run on QIIME processed sequence data to identify the “core” microbiome and diversity analysis. Data were filtered for low count and low variance using default parameters, which resulted in the removal of 132 low abundance features based on prevalence and 126 low variance features based on inter-quantile range (iqr). The “core” microbiome refers to the set of genus level taxa that were detected in a high fraction across the tested soils using the following threshold levels: Sample prevalence (50%) and relative abundance of 0.2%. A total of 1128 features remained for downstream processing after data filtering step. Libraries were then rarefied to the minimum library size and total sum scaling was performed to account for sample variability such that biologically meaningful comparisons can be drawn. Groups shown in the figures refer to the metadata, in which soils used in DC and MT experiments were labeled as group 1, along with the following groupings: Group 2 (DC Gen1), group 3 (DC Gen2), group 4 (MT Gen1), and group 5 (MT Gen2), respectively. Dendrogram analysis was run at the genus level using the Bray–Curtis index and selecting the experimental factor as grouped in the metadata file. Further ordination analysis on the amplicon-based metagenomics data was performed at the genus level using the MicrobiomeAnalyst built-in tools, such as α diversity (Chao1 measure with T-test/ANOVA), β diversity plotted as PCoA (Bray–Curtis distance method with PERMANOVA), univariate analysis using T-test/ANOVA with an adjusted cutoff value of 0.05, and differential abundance analysis at the genus level, calculated using EdgeR at an adjusted *p*-value cut off of 0.05.

### 2.4. Identification of the Isolated Strains

DNA isolation from the DC and MT gels and from the bacterial and fungal strains isolated from this study was performed using DNeasy PowerLyzer Microbial Kit (Qiagen Inc., Germantown, MD, USA). Bacterial and fungal strains were identified based on the 16S and 18S rDNA sequencing analysis, respectively. PCR was performed with 27F-1492R universal bacterial primers [48] at an initial denaturing step at 95 °C for 3 min, followed by 35 cycles of denaturation at 94 °C for 40 s, annealing at 55 °C for 30 s, extension at 72 °C for 60 s, and a final extension step of 72 °C for 5 min. Fungal PCR was performed with FR1/NS1 universal fungal primers [49] at an initial denaturing step at 95 °C for 8 min, followed by 35 cycles of denaturation at 95 °C for 30 s, annealing at 47 °C for 45 s, extension at 72 °C for 60 s, and a final extension step of 72 °C for 10 min. The sequences obtained were identified for taxonomy using NCBI BLAST.

### 2.5. Evaluation of Uranium Resistance of the Isolated Strains

Growth efficiencies of the isolated bacterial strains on U were determined as follows. Briefly, LB media was mixed with uranyl nitrate to yield different concentrations of U, ranging from 0 to 5 mM. The isolated strains grown overnight in LB media were then added to obtain an OD_600_ of ~0.3. The OD was measured every 4 hours for 3 days at 28 °C with constant shaking. All the assays were performed in triplicate. Growth efficiencies of the isolated fungal strains on U were determined by growing the strains on potato dextrose (PD) agar in the presence of different concentrations of U (0–15 mM). Two-day old fungal cultures from PD broth were inoculated in the center of PD agar plates. The resulting diameter of the fungal colony was measured every 24 hours at 28 °C as a measure of resistance to U.

### 2.6. Evaluation of Uranium Depletion by the Isolated Microbiota Using Soil Microcosms

Uranium reduction by the isolated strains was evaluated in soil microcosms. In a 250 ml Erlenmeyer flask, 10 g SRS uraniferous soil was added to liquid growth media (LB for bacteria and Potato Dextrose Broth (PDB) for fungi) to make a total volume of 100 ml and autoclaved. Uranium was added to the flasks, as uranyl nitrate, to reach a final concentration of 1 mM or 2 mM U. All microcosms were set up in duplicates. The bacterial strains grown overnight in LB media were used as bacterial inoculum, the fungal strains grown for 3 days were used as fungal inoculum. A total of 1ml of each inoculum was added to the respective microcosm and incubated at 30 °C at 100 rpm for 96 hrs. Samples to determine U concentration were withdrawn every 24 hrs. Each sample was centrifuged at 7000 rpm for 10 min and supernatant was collected and acidified to 2% HNO_3_ (v/v). These samples were analyzed using an ICP-OES (Perklin Elmer, Waltham, MA, USA) for the presence of soluble U.

### 2.7. Genome Sequencing and Bioinformatic Characterization of Strain MT2

Genomic DNA from strain MT2 was extracted using a MoBio PowerLyzer kit (Qiagen Inc., Germantown, MD), prepared for sequencing with Illumina Nextera XT library preparation protocol (Illumina Inc., San Diego, CA, USA), and sequenced using a NextSeq500 instrument. De novo assembly was performed using the Spades assembler [50] on raw Illumina reads, with multiple k-mers specified as “-k 27,47,67,87,107,127”. Coverage levels were assessed by mapping raw Illumina reads back to the contigs with the MEM algorithm of Burrows-Wheeler Aligner software [51] and computing the number of reads aligning per contig times the length of each read and divided by the length of the contig. Contigs were filtered by coverage by first assessing the relationship between cumulative assembly length and coverage, over coverage-sorted contigs. We took 50% of the coverage level at half the total assembly length as a coverage threshold and removed contigs with coverage less than this value. Automated gene annotations were generated using Augustus [52], using a *Penicillium oxalicum strain* (KB644408.1) for training. A circular genomic map of strain MT2 was generated using the CGView Comparison Tool pipeline or the embedded tool in the EDGAR pipeline [53].

Based on the presence of coding sequences for genes Tsr1, Cct8, RPB1, and RPB2A within the genome sequence of strain MT2, a phylogenetic tree was constructed. Sequences were aligned using the Muscle algorithm in the MEGA X package [54], realigned by codon, concatenated, and used in the phylogenetic analysis. The evolutionary history was inferred by using the Maximum Likelihood method and Tamura–Nei model [55]. The tree with the highest log likelihood (−49122.85) is shown and the percentage of trees (100 iterations) in which the associated taxa clustered together is shown next to the branches. Initial tree(s) for the heuristic search were obtained automatically by applying Neighbor-Join and BioNJ algorithms to a matrix of pairwise distances, estimated using the Maximum Composite Likelihood (MCL) approach, and then selecting the topology with superior log likelihood value. A discrete γ distribution was used to model evolutionary rate differences among sites (5 categories (+G, parameter = 1.1746)). The rate variation model allowed for some sites to be evolutionarily invariable ([+I], 29.13% sites). The tree is drawn to scale, with branch lengths measured in the number of substitutions per site. This analysis involved 11 nucleotide sequences. Codon positions included were 1^st^ + 2^nd^ + 3^rd^. There was a total of 9777 positions in the final dataset.

### 2.8. Comparative Genomics of Strain MT2

Further evolutionary relatedness of strain MT2 was inferred by using EDGAR [53], which aided in the identification of four closest taxonomic relatives of strain MT2, which were chosen for further analysis. The average nucleotide identity (ANI) of strain MT2 was estimated between strain MT2 and the closest phylogenetic neighbor, *Penicillium janthinellum* strain NCIM1366, as previously described [56]. This ANI calculator utilizes the OrthoANIu algorithm, which is an improvisation of the original OrthoANI algorithm, which uses USEARCH instead of BLAST. Data derived from ANI was also augmented by assessment of a potential novelty of strain MT2 using digital DNA-DNA hybridization (dDDH) through the Genome-to-Genome Distance web service [57].

### 2.9. Nucleotide Sequence Accession Number

The 16S rDNA sequences of strains isolated in this study are deposited in NCBI GenBank, as shown in parentheses, as follows: *Burkholderia* sp. DC2 (MH595922), *Burkholderia* sp. DC3 (MH595923), *Penicillium* sp. DC1 (MH595924), *Penicillium* sp. MT1 (MH595925), and *Penicillium* sp. MT2 (MK300050), respectively.

### 2.10. Metagenomic Sequence Accession Number

The metagenomic sequences obtained from this study are available from NCBI’s Sequence Read Archive/European Nucleotide Archive, accession number SUB4896106 under BioProject PRJNA509237 [58].

### 2.11. Whole Genome Sequence Submission

The Whole Genome Shotgun sequence of *Penicillium* sp. strain MT2 reported in this study has been deposited at DDBJ/ENA/GenBank and is available using accession number PZKB00000000 [59].

## 3. Results and Discussion

### 3.1. Changes in the Bacterial, Archaeal, and Fungal Assemblages between DC and MT Generations

Shotgun metagenomics was performed to evaluate the soil microbiome colonizing the DC and MT chamber niche relative to the soils used for these experiments. The total sequence read counts obtained from all the soil samples used in this study are shown in Appendix A; ~20% of contigs across the soils and DC/MT were annotated against taxonomy and/or gene functions. MicrobiomeAnalyst pipeline binned the sequences into 1424 OTUs, which were then used for data comparisons, visualizations, and statistical analyses.

The uraniferous soils used in this study mainly consisted of 10 phyla with greater than 50% of total relative abundance (Figure 1A). Among these, *Proteobacteria* accounted for almost half of the diversity, followed by *Actinobacteria*, *Acidobacteria*, and *Firmicutes*. Similar results have been observed in a U contaminated soil in southern China, which also revealed *Proteobacteria* and *Actinobacteria* predominated as a function of U contamination [40]. Furthermore, 16S identification of a U-contaminated soil microbial community of Limousin, France, and several other studies, also show preponderance of *Proteobacteria* and *Acidobacteria*, with the members of genus *Geobacteraceae* dominating the overall bacterial population [60,61,62]. Proteobacteria phyla and the *Geobacteria* genus are both well known to be metabolically, ecologically, and environmentally diverse and, hence, outcompete other phyla in a variety of soil conditions, including contamination with U. Of further interest was the observation that Proteobacterial communities significantly proliferated in the growth conditions presented by both DC and MT (Figure 1A). Conversely, the relative abundance of *Actinobacteria*, which was higher in the soils, reduced in both the generations of DC and MT. Similarly, relative abundances of *Acidobacteria* and *Firmicutes* also reduced in the second generation of both DC and MT, when compared to the soils.

At the genus level, *Burkholderia* was identified to be most abundant in the SRS uraniferous soils (Figure 1B), which was also identified as the “core” microbiome (Appendix A), regardless of the DC/MT treatment. Differences in the *Burkholderia* spp. between soils and DC/MT samples were further assessed using univariate statistical comparisons and differential count analysis using EdgeR (Appendix A). Taking the original counts as well as log transformed data, we observed differences between soils and DC/MT samples, such that the abundances of *Burkholderia* spp. significantly increased in both DC and MT generations (Appendix A), especially in Gen 1 compared with soils, thus providing unequivocal evidence that DC/MT provided specific culture conditions that enhanced specific bacterial communities to further thrive in the presence of environmentally-relevant nutrient concentrations being provided to the microbiota via the membranes separating the soil from the gel-embedded samples. Other dominant bacterial genera identified in this study included *Rhodanobacter* spp., *Bradyrhizobium* spp., *Pseudomonas* spp., and *Dyella* spp., respectively. These bacterial groups have also been reported to survive in and colonize uraniferous soils, including *Rhodanobacter* [63], *Pseudomonas* spp. [64], and *Dyella* spp. [39], which is in line with findings from this study.

As stated above, the “core” microbiome refers to the set of genus level taxa detected in a high fraction across the tested soils using the following threshold levels: Sample prevalence (50%) and relative abundance of 0.2%, respectively. The core microbiome abundance further increased significantly in the subsequent transfers, such that *Burkholderia* spp. outcompeted all other genera and proliferated to as high as 60%–70% of the total abundances in both Gen1 and Gen2, respectively (Figure 1B). However, the relative abundance of *Rhodanobacter* and *Bradyrhizobium* was not significantly different in the DC and MT gels when compared to the soils. It is noteworthy that our previous studies have also found the dominance of *Burkholderia* spp. in U-contaminated SRS soils [34,37]. In fact, another group working on subsurface soils of the DOE Old Rifle Processing Site in Colorado, which has undergone a similar historical contamination with U as the SRS site, found Burkholderiales to comprise approximately 30% of the total subsurface microbial community [65,66]. Similarly, from other DOE sites, such as in Oak Ridge, Tennessee and Rifle, CO, *Burkholderia*-like microorganisms have been demonstrated to comprise a significant proportion of the U reducing microbiota [66,67,68,69]. Moreover, the β-proteobacterial *Burkholderia* genera are not only found in uranium-rich soils, but occur in diverse ecological niches ranging from soils to aqueous environments and many *Burkholderia spp.* are symbionts of both plants and animals [70]. The widespread occurrence of *Burkholderia* spp. in contaminated environments is linked to their rigorous biodegradative and metal resistance abilities [71,72,73,74,75], including to uranium [65]. Therefore, it appears that *Burkholderia* spp. possess a unique ability to not only survive in uraniferous soils, but also likely play a critical role in the microbially-mediated remediation of U within the SRS soils.

Furthermore, dendrogram analysis revealed that the soil microbiome community was distinctly different between soils and the DC/MT generations (Appendix A), suggesting that certain soil bacterial communities were favored in the conditions presented by DC/MT. Notably, bacterial communities that developed in Gen1 DC and MT clustered together and away from Gen 2 DC and MT, suggesting that incubation times influenced growth of bacterial diversity and not so much as the membrane size difference between DC/MT. To further evaluate the statistical differences between soils and DC/MT microbiome, α and β diversity at the genus levels were estimated and ordination plotted as PCoA using the Bray–Curtis index (Figure 2). The α diversity, which is a measure of the diversity within each sample, was observed to be highest for soils relative to the DC and MT generations, suggesting that growth of certain microbiota was favored by the culture conditions. Specifically, α diversity decreased significantly in both DC Gen1 and MT Gen 1, suggesting that some bacterial groups flourished and outcompeted those that were stressed under the DC/MT conditions (Figure 2A). This can also reflect the initial stress posed by the artificial growth conditions being presented by DC/MT. However, the α diversity almost reached back up to the level of soils in both DC/MT Gen 2, suggesting that the soil microbiota is potentially resilient and overcame the initial stress of the artificial growth conditions, as was observed by reduction of α diversity in Gen1 experiments. The β diversity, which is a measure of the diversity between the evaluated samples, showed that microbiota in the soils used for establishing the DC/MT experiments were not similar but switched to become similar as a function of incubation time or generations, but not as a function of DC and MT treatments. This suggests that both DC and MT growth conditions permitted for similar microbial growth, fed by the nutrients that passively diffuse through the DC/MT membranes into the chamber plugs.

Although not the focus of this study, we also evaluated the archaeal diversity between the soils and the diffusion chamber/microbial trap experiments. Similar to bacterial metagenomic analyses, overall, the archaeal analysis showed Euryarchaeota and Thaumarchaeota phyla to dominate the soils and, to a lesser extent, Crenarchaeota was also present (Appendix A). As a function of DC and MT growth conditions, Euryarchaeota phyla increased in the generations. Conversely, the phyla Thaumarchaeota was present as the most abundant in soil samples used to establish the DC and MT experiments, but its relative abundance declined in the DC and MT samples. As shown in the core microbiome analysis, the archaeal genera that dominated include *Methanosarcina*, *Nitrososphaera*, *Methanocaldococcus* and *Methanocella* (Appendix A). It was observed that the relative abundances of these genera declined as a function of DC/MT generations, which indicates that anaerobic conditions were not maintained, for the most part in our DC/MT cultures, so that conducive conditions remain persistent for the growth of bacterial communities, which was the main aim of this study.

Similar to the bacterial identification, archaeal core members of the soil community mainly consisted of *Methanosarcina* spp. (Appendix A), followed by *Nitrosphaera* and *Methanocella* spp. Notably, for the first time, *Methanosarcina* spp. were recently demonstrated to influence U speciation by coupling oxidation of acetate to the reduction of U(VI) [76] in the Rifle 24 acre experimental site, located in close proximity to the Colorado River, an abandoned U ore processing facility. It is very likely that *Methanosarcina* spp. play critical roles within the SRS U-contaminated soils, especially under anaerobic conditions. Previous studies of acetate-stimulated U bioremediation have largely focused on the *Geobacter* spp. because no other microorganism capable of coupling oxidation of acetate with U(VI) reduction has been demonstrated thus far. We are tempted to speculate that, in the SRS soil habitat, *Burkholderia* spp. are active in the surficial aerobic soil niches and, once the redox potential becomes anoxic, the U bioremediation is catalyzed by *Methanosarcina* spp., which will be the subject of our future research. Furthermore, dendrogram analysis showed that the soil archaeal community was distinctly different relative to the DC/MT generations (Appendix A), suggesting that certain archaeal communities preferred the DC/MT conditions and thus thrived. Archaeal communities that developed in Gen1 DC and MT clustered separate to the Gen2 and soils, indicating that archaea were acclimatizing in the Gen 1 conditions. Like the bacterial communities, the overall trend was that the incubation times influenced growth of archaeal diversity and not as much as the membrane size difference between DC-MT.

When fungal community structure was evaluated, the Ascomycota phylum was present as much as 75% of the total relative abundance in the tested soils, which increased to almost a 100% within the DC/MT plugs (Figure 3A). The phyla Basidiomycota was observed as the second most abundant phylum in the soils, but its abundance reduced significantly in the conditions presented by both DC and MT, respectively. The phyla Glomeromycota and Chytridiomycota were present in very low abundance in the soils. These phyla completely disappeared in the DC and MT plugs. As it can be seen in Figure 3B, the soils harbored a significant diversity of fungal genera. However, genera belonging to the Ascomycota phylum, such as *Penicillium, Aspergillus*, *Talaromyces*, and *Fusarium*, were the most predominant. It is interesting to note that the relative abundance of *Penicillium* spp. increased considerably in both DC and MT, relative to the soils. *Penicillium* spp. reached a relative abundance of 40%–48% in the 2^nd^ MT generation. Similar response was shown by *Aspergillus* spp., whose relative concentration significantly increased in both the DC and MT plugs. From these results it can be concluded that the DC and MT gels are selective for *Penicillium* and *Aspergillus* spp. It is noteworthy that similar to *Burkholderia* spp., which was one of the most abundant bacterial genera in the soils and also enhanced in the culture conditions presented by the DC and MT plugs, the fungal *Penicillium* genus was also facilitated to thrive by the in situ DC/MT culture conditions. Thus, it appears that those communities having the ability to resist and even bioremediate U were preferentially selected by the DC/MT technique. This observation is supported by the isolation and screening of both *Burkholderia* and *Penicillium* groups of soil microbiota in the presence of uranium, as shown later in this study.

### 3.2. Gene Functional Analysis in Soils Relative to DC and MT Chambers

We also investigated whether bacterial, archaeal, and fungal community shifts also reflected in functional shifts that were being performed in both the soil habitat and the DC/MT growth conditions, respectively. As shown in Figure 4A,B, shotgun metagenomics at the subsystem level 1 indicated that genes performing functions related with carbohydrate metabolism, virulence, and respiration were the most abundant functions performed across the soils and DC/MT culture conditions. Albeit at lower levels, functions related with stress response, membrane transport, and metabolism of aromatic compounds were also observed across the evaluated SRS soils. Some of these findings are in line with previous studies, where carbohydrate metabolism and membrane transport functions were overrepresented in soils contaminated with U [40], indicating that U stress enhances the need for the soil microbiota to utilize carbohydrates, which may also drive up the respiration rates. In context with membrane transport, it has been previously shown that membrane transport is a critical feature for membrane-dependent uranium transport by the soil-borne bacterial cells and, in doing so, intracellular metabolism and energy transduction are optimally maintained for the uranium-resistant microbiota [15,38,40].

Additionally, of major interest was the identification of phosphate metabolism gene functions across the SRS soils used in this study (Figure 4B). However, no significant differences were observed for the phosphatase function between soils and DC/MT culture conditions, suggesting that environmental microbiomes have remarkable functional redundancy, as also reported previously [77]. Therefore, despite the observed community structure shifts between the soils and DC/MT culture conditions, functional processes likely remained constant. Regardless, the phosphate functional process in U-contaminated soils remains to be of significant interest because U biomineralization via bacterially-mediated phosphatases is a highly effective mechanism to convert U into a U-mineral [78,79]. Typically, bioprecipitation of metals occurs via acid or alkaline phosphatases cleaving an organic phosphate compound to free up the phosphate moiety, which then precipitates heavy metals from the solution phase to generate uranyl phosphate precipitate, a highly insoluble mineral facilitating long-term in situ U immobilization. This phosphatase-mediated bioprecipitation strategy is very promising for treating nuclear waste and heavy metals and, thus, better stewardship of contaminated habitats.

It was noteworthy that at subsystem level 3, several gene classes known to facilitate the bioremediation of contaminants were also identified, including the oxidative stress-related proteins of the YgfZ family, the YgfZ-Fe-S family, and phosphate metabolism (Figure 4C). It has been reported that the YgfZ proteins participate in assembly or repair of iron/sulfur clusters that perform a large number of conserved cellular processes such as catalysis, electron transport, gene regulation, DNA replication and repair, and central metabolism [80,81].

When heatmaps were plotted to tease out the gene functions that potentially shifted between the soils and DC/MT samples, it became apparent that a cohort of functions were unique to the soils (Appendix A), such as anaerobic degradation of aromatic compounds, coenzyme F420, and catabolism of unknown compounds. Conversely, functions such as stress response, chemotaxis, transport proteins, quorum sensing, and biofilm formation were overrepresented in the DC/MT conditions. Principal component analysis (PCA) of this data (Appendix A) clearly showed that soil gene functions clustered together, but not with the DC/MT generations, meaning that gene functional differences were largely driven by the growth conditions presented by DC/MT and not so much by the nature of DC/MT filter sizes used in establishing the chambers, a similar trend shown by the β diversity analysis on the taxonomic assemblages. Dendrogram analysis further supported these observations such that soils used for DC/MT clustered together and away from generation 1, which clustered away from the generation 2 gene functions, as shown in Appendix A. Overall, the gene functional analysis revealed that functions within the DC/MT chambers did change relative to the soils, but these are to be expected because of an artificial environment posed by the DC/MT culture conditions. Furthermore, no specific functions related to U cycling or speciation were identifiable in both soils and DC/MT samples, which indicated that the soil microbiota has adapted to the presence of U stress since the SRS soils have continued to contain elevated U levels since the 1950s.

### 3.3. Isolation of Bacteria and Fungi from DC/MT Chambers

Using conventional enrichment cultures, our laboratory has previously isolated *Arthrobacter* spp. and *Burkholderia* spp., which have shown to resist high concentrations of U [33,34]. To augment the metagenomics-based findings reported herein, we performed axenic culture studies. Specifically, bacterial and fungal strains were isolated from the DC and MT gel plugs. The isolations were performed from generation 2 of the DC and MT passage by dilution of the retrieved plugs and spread-plating onto LB/PD agar supplemented with 2 mM U, as well as soil extract media. From the DC approach, two noteworthy bacterial isolates (DC2 and DC3) and one fungal isolate (DC1), were retrieved. The MT approach, which is used for the isolation of actinomycetes, led to the isolation of two fungal strains (MT1 and MT2). However, the isolates were obtained only on LB/PDA media, but none on the soil extract media, which was unexpected as the DC/MT plugs are continuously provided with environmentally relevant concentrations of nutrients from the underlying soil. Despite prolonged incubations (~3 weeks) of the soil extract plates, no bacterial or fungal colonies could be retrieved from the DC/MT plugs, which is in contrast to the studies by Bollmann et al., who were successful at isolation of several uncultivable bacteria from uraniferous soils on soil extract media [7].

The isolated strains were identified by 16S/18S gene sequencing as *Burkholderia* spp. and *Penicillium* spp., respectively. Phylogenetic trees of the isolated *Burkholderia* spp. are shown in Appendix A and those of *Penicillium* spp. are shown in Appendix A. This clearly showed that the DC/MT facilitated the in-situ growth of those bacteria and fungi that were dominant in the SRS soils and most likely perform an environmentally relevant service within the studied soils, such as uranium bioremediation. Moreover, our previous studies [38] and other researchers [66,67,68,69] have also found *Burkholderia* to be the predominant species in U contaminated sediments. Further, *Penicillium* spp. has also been shown to resist a variety of heavy metals, including U, especially at high concentrations [20]. This conclusively demonstrated the successful use of diffusion chambers and microbial traps to isolate environmental microbiota with desirable traits. In our study, this trait was the isolation of U-resistant microbiota. This approach also circumvents typical isolation strategies on high carbon and other nutrients that are not reflective of environmental conditions because the biomass colonizing within the plug is constantly fed by environmentally-relevant nutrients diffusing into the chamber from the soil slurry beneath the chamber.

### 3.4. Screening for Uranium Resistance in the Isolated Strains

To evaluate the growth tolerance of isolated bacterial strains, they were grown in LB media amended with U ranging from 0–5 mM concentrations. As shown in Figure 5, *Burkholderia* spp., DC2, and DC3 grew well up to 2 mM U (to reduce clutter, only 3 U concentrations are shown on the graph). On further increase in the concentration of U to 2.5 mM, both the strains failed to grow (Figure 5A,B).

The fungal strains were grown in PD agar, amended with variable concentrations of U (0–15 mM), for 4 days and the diameter of the fungal colonies was measured every 24 hours (Figure 5C–E). It was observed that all the *Penicillium* spp., DC1, MT1, and MT2, grew up to 10 mM U concentration and failed to grow at higher U concentrations (15 mM). It was also observed that *Penicillium* spp. MT1 and MT2 were more resistant to U than *Penicillium* spp. DC1, as it grew at a much faster rate at higher U concentrations of up to 10 mM, indicating that these strains are robust in context with U resistance and/or bioremediation.

### 3.5. Uranium Depletion by the Isolated Microbiota in Soil Microcosms

To evaluate U depletion by the isolated microbes, soil microcosms were set up at a U concentration of 1000 µM and 2000 µM. All the three fungal isolates were able to deplete approximately 90% of soluble uranium in 4 days (Figure 6A), at a 2 mM U concentration. Specifically, *Penicillium* sp. DC1, MT1, and MT2 were able to deplete 94%, 89%, and 91% of soluble U, respectively. *Burkholderia* sp. DC2 and DC3 depleted 54% and 96% U, respectively (Figure 6A). Figure 6B depicts the U present in soil microcosms for *Penicillium* sp. MT2 over 4 days. Within the first day, this strain depleted most of the spiked U from the soluble phase, indicating the strong potential of this fungal isolate for U bioremediation in the SRS soils.

### 3.6. Genome-Centric Evaluation and Comparative Genomics of Strain MT2

The whole genome sequence of strain MT was found to be approximately of 34,123,110 bp in size, with a G+C content of 51.15%, and consisted of a total of 11,470 genes. Strain MT2’s whole genome sequence contained an N_50_ length of 179,326 bases, with the longest contig length of 606,913; a circular genome map is shown in Figure 7, drawn relative to several randomly selected Penicillia. To infer taxonomic affiliation of strain MT2, a phylogenetic tree based on manually queried housekeeping genes Tsr1, Cct8, RPB1, and RPB2A, from the genome sequence of strain MT, is shown in Figure 8A. A similar approach was recently used to characterize a novel *Penicillium* spp., isolated from an extremely metal-rich mining site in Russia [82]. This analysis revealed the closest affiliation of strain MT2 with *Penicillium janthinellum* strain NCIMI366. Further iteration of the evolutionary relatedness of MT2 genome-wide comparisons were run with a cohort of 28 other *Penicillium* genomes, which confirmed that the closest taxonomic affiliation of MT2 was with *P*. *janthinellum*, followed by *P*. *decumbens* strain IBT, *P*. *oxalicum,* and *P*. *subrubescens*, respectively (Figure 8B). Based on these analyses, it appears that MT2 is a potentially new species related to *P*. *janthinellum.*

Further bioinformatic evidence garnered to support this claim is based on the Average Nucleotide Identity (ANI) value of 91.45% between strain MT2 and *Penicillium janthinellum* strain NCIM1366. Note that, typically, the ANI values between genomes of the same species are above 95% [83]. This is highly suggestive that MT2 is a new *Penicillium* species. Similar to the ANI, when digital DNA-DNA hybridization (dDDH) was calculated, the probability of strain MT2 being the same species as *P*. *janthinellum* strain NCIM1366 was estimated to be merely 6.05% (via logistic regression). Note that a DDH > 70% (via logistic regression) would be inferred as the two genomes belonging to the same species. When syntenic association of MT2 with *P*. *janthinellum* was evaluated, an extremely low similarity was seen (Appendix A), further supporting the observation that MT2 is a new *Penicillium* spp., isolated from the microbial trapping approach. Syntenic regions are defined as homologous multi-gene regions in two or more genomes in which a repertoire of genes are conserved, along with the potential conservation of transcription direction and linear gene order. Thus, this synteny-based evaluation strongly suggests that MT2 consists of a mosaic of Penicillia genes, but distinctly divergent from other described species of this genus. Taken together, this is strong comparative genomics and bioinformatics-based evidence that strain MT2 has a unique taxonomic position relative to other Penicillia members evaluated in this study and is most likely a new species.

Of further interest was EDGAR pipeline-based comparative genomics with the above stated genomes of Penicillia, which identified 1904 genes (16.5% of the total genome) that were unique to MT2, relative to the four closest relatives, as shown in Figure 9. When these unique genes were further analyzed, a suite of heavy metal and drug resistance determinants were identified in strain MT2, such as ABC transporter proteins, cytochromes, and a variety of efflux pumps (Appendix A). This comparative genomic analysis confirms the strong catabolic and bioremediation potential possessed by the novel MT2 strain, as also supported by the soil microcosm studies (Figure 6).

In summation, the ability of bacterial strains to survive in uraniferous ecosystems has been well demonstrated for anaerobic ecosystems, but the potential of aerobic microorganisms to biomineralize uranium has been largely ignored and, hence, remains understudied. Overall, this study reports on an innovative metagenomics-guided isolation of aerobic bacteria and fungi that resist high concentrations of U. Notably, we first compared the microbial diversity using shotgun-based metagenomics in soil samples collected from the SRS and the microbiota colonizing within the DC and MT gel plugs across 2 different incubation times (i.e., generations or Gen). Simultaneously, we isolated U resistant bacteria and fungi directly from soils as well as the Gen1 and Gen2 agar plugs. *Burkholderia* spp. and *Penicillium* spp. were the isolated bacterial and fungal strains regardless of being treated as DC or MT, respectively. Of major mention is the isolation of a potentially new species of Penicillia using the microbial trap approach, as supported by comparative genomic analysis. The metagenomics data was in line with the culture-based findings such that the top 3 bacterial genera are taxonomically affiliated with *Burkholderia*, *Rhodanobacter,* and *Bradyrhizobium* in the soils. This increased to as much as 60%–80% under the conditions of DC and MT. Similarly, the top 3 fungal genera were identified as *Penicillium*, *Aspergillus,* and *Talaromyces,* but *Penicillium* colonized more favorably under the DC and MT conditions. A relative abundance of genes, coding for different functions, were also identified, which mainly belonged to carbohydrate metabolism, virulence, respiration, and, at a finer scale (subsystem level 3), the gene families for DNA replication, flagellum, and YgfZ family were the major ones, but not much change was observed between soils and the DC/MT conditions. With this study, we provide the proof-of-concept for utilizing the diffusion chamber and microbial trapping techniques for obtaining a broader understanding on the bacterial, archaeal and fungal communities that persists in a U contaminated soil habitat, as well as isolation of the key microbial players for further “omics”-based downstream analysis.

## Figures and Tables

**Figure 1 genes-10-00325-f001:**
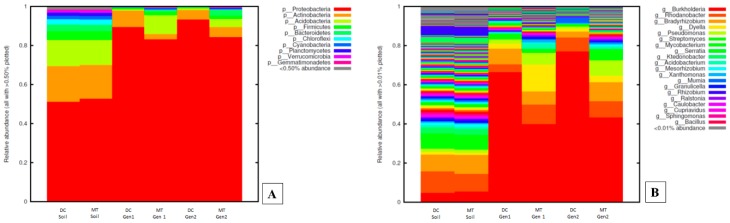
Bacterial diversity plotted as relative abundances shown at the phyla (**A**) and genus (**B**) levels, identified from the soils and the DC/MT chambers, respectively.

**Figure 2 genes-10-00325-f002:**
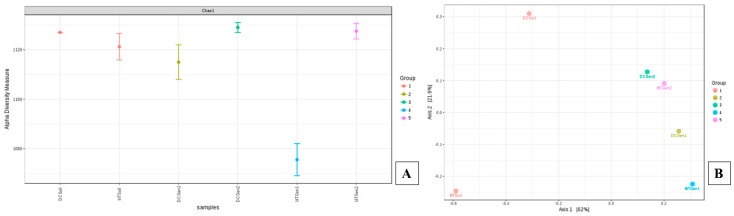
Shown are the α (**A**) and β diversity (**B**) analysis and significance testing between the soils and DC/MT chambers. A [PERMANOVA] R-squared: 0.9976; *p*-value < 0.066667 was obtained from this analysis. Groups shown in the figure refer to the metadata in which soils used in DC and MT experiments were binned, as follows: Group 1 (soils used for DC/MT), group 2 (DC Gen1), group 3 (DC Gen2), group 4 (MT Gen1), and group 5 (MT Gen2), respectively.

**Figure 3 genes-10-00325-f003:**
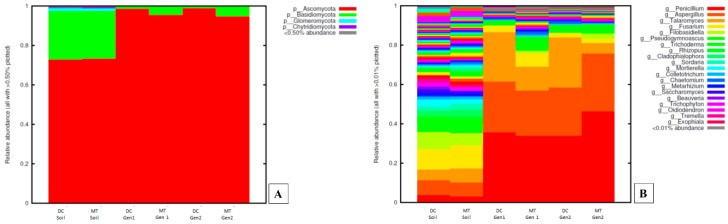
Fungal diversity plotted as relative abundances shown at the phyla (**A**) and genus (**B**) levels, identified from the soils and DC/MT chambers, respectively.

**Figure 4 genes-10-00325-f004:**
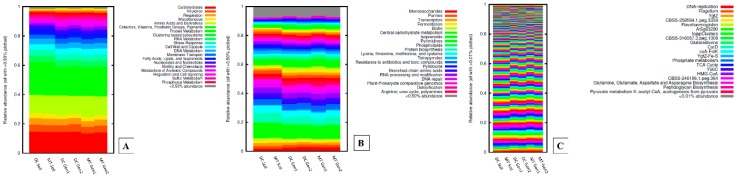
Gene functions identified from the SRS soils relative to different generations of DC/MT binned at subsystem levels 1 (**A**), 2 (**B**), and 3 (**C**), respectively.

**Figure 5 genes-10-00325-f005:**
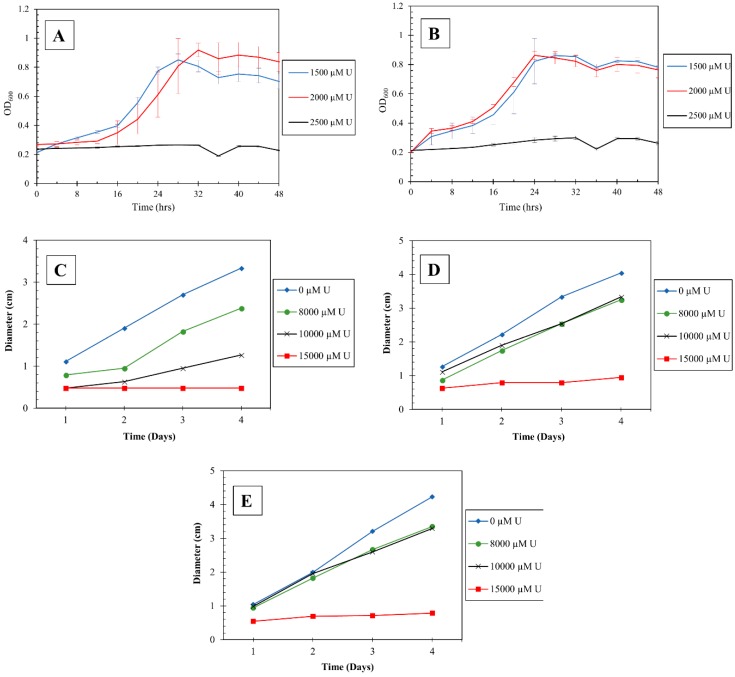
Shown are the growth responses of *Burkholderia* spp. strain DC2 (**A**) and DC3 (**B**), at different concentrations of uranium. Also shown are the growth responses of *Penicillium* spp. DC1 (**C**), MT1 (**D**) and MT2 (**E**) at different concentrations of uranium.

**Figure 6 genes-10-00325-f006:**
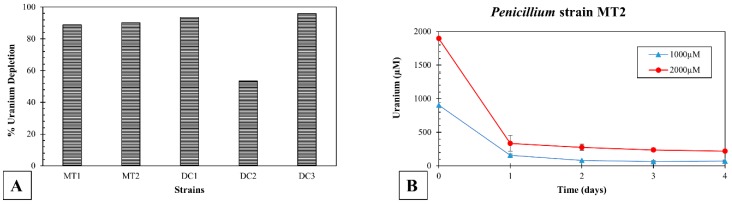
Soil microcosms established with uranium (U) (1000 µM and 2000 µM) and the bacterial/fungal isolates obtained from DC and MT chambers. Shown is the percent depletion of soluble U at the experiment cessation time point for all strains, at 2000 µM (**A**), and U depletion over a 4-day period by strain MT2 (**B**), respectively.

**Figure 7 genes-10-00325-f007:**
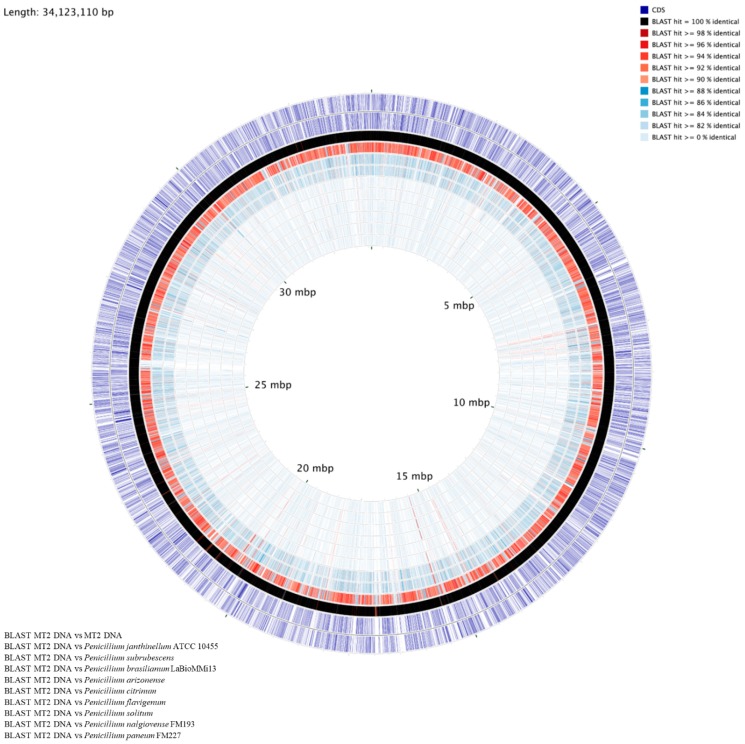
Shown is a circular genomic map of *Penicillium* sp. strain MT2. The outer two rings show the positions of CDS features on the forward and reverse strands, respectively. The remaining rings display the results of blastn comparisons between the MT2 genome and several comparison genomes, as follows: MT2, *Penicillium janthinellum* strain NCIM1366, *Penicillium subrubescens* strain CBS 132785, *Penicillium brasilianum*, *Penicillium arizonense* strain CBS 141311, *Penicillium citrinum*, *Penicillium flavigenum* strain IBT 14082, *Penicillium solitum* strain RS1, *Penicillium nalgiovense* strain IBT 13039, and *Penicillium paneum* FM227, respectively. For the blast comparisons, an E-value cutoff of 0.000001 was used. The map was generated using the CGView Comparison Tool, available to download from: http://stothard.afns.ualberta.ca/downloads/CCT/.

**Figure 8 genes-10-00325-f008:**
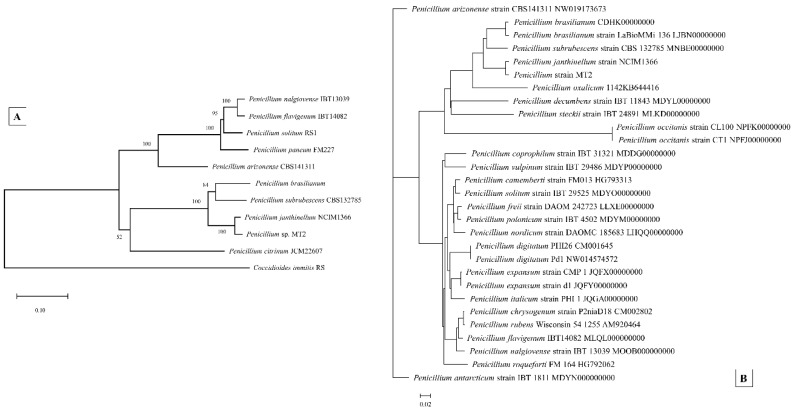
Shown is a phylogenetic tree constructed with maximum likelihood using concatenate sequences of the housekeeping genes Tsr1, Cct8, RPB1, and RPB2 (**A**), and (**B**), a genome-wide phylogenetic tree constructed with a cohort of 28 other sequenced *Penicillium* species available via the NCBI genome submission portal.

**Figure 9 genes-10-00325-f009:**
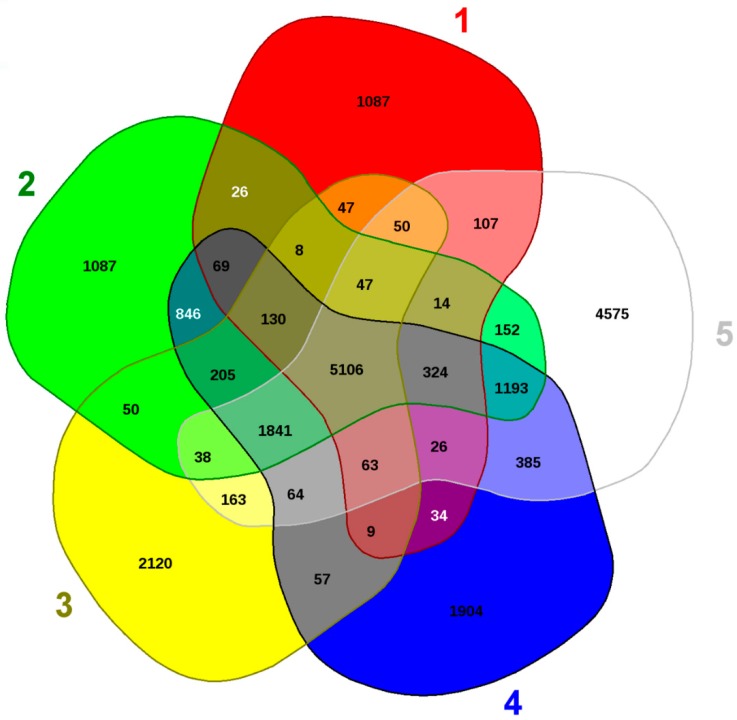
Whole genome sequence-based Venn diagram generated between *Penicillium* sp. strain MT2 with four closest taxonomic relatives. Venn diagram sectors belong to 1, *Penicillium decumbens* strain IBT; 2, *Penicillium janthinellum* strain NCIMI366; 3, *Penicillium oxalicum*; 4, *Penicillium* sp. strain MT2; and 5, *Penicillium subrubescens* strain CBS, respectively. The number of singleton genes appear in red, green, yellow, blue, and white areas for strains 1–5 listed above, along with their core genomes (centered gray area).

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
