# Peer review of "Metagenomics-Guided Survey, Isolation, and Characterization of Uranium Resistant Microbiota from the Savannah River Site, USA"

_genes, 2019, doi:10.3390/genes10050325_

Round 1
Reviewer 1 Report
Manuscript# -genes-473594,
Metagenomics‐guided Analysis, Isolation and Characterization of Uranium Resistant Microbiota Using a High‐throughput Approach
Merits:
After long the reviewer came across the manuscript with bench work data backed up by the bioinformatics data and vice versa. This effort to bring the two world together is appreciated
Improvements:
Minor:
· Title of the manuscript is bit misleading and has to be changed, culture-omics is taking the lead here, and then the body of the experiment is framed: the author should address this in the title
· Line 67 – 88: Author can write this para in a balanced manner explaining the comments at family or genus level phyla is a bit too vague.
· Line 199 – 215: It's always good to have a simple diagram while explaining material method less know to the public or write it in a language which is conceivable in one read. Reviewer suggest the authors do either of the two
· Line 216: Microbiome analysis: If Author has used two different set of program one for total metagenome (MEGAN) and other which is used for amplicon analysis (Qiime) review expect the authors to explain the transition and valid bash/programming codes should be provided with the manuscript.
· Line 252: ……..on 16S or 18S rDNA sequencing analysis. :
The statement is very misleading; please use the scientific language to justify your case, be specific of what was used for the sequencing analysis.
· Line 297: What was the concatenation strategy please provide the relevant codes or reference.
Major Improvement:
· All and every reference of MT2 being novel has not been stated as per the rules laid by the taxonomic society. Please refer to IJSEM fungal novel species reference to execute the same. Using of ANI and dDDH as a speciation marker is only valid for Prokaryotes not for Eukaryotes. Eukaryote taxonomy does not follow those standards, for example, Human, Baboon, and Chimps share more than 98.6% of genome yet they are different species. Reviewer request authors to correct this throughout the manuscript and validate MT2 as new species as per the defined rules.
· Line 475 – 525: Reviewer would like to stress here that the author have conducted a meta-genomic analysis and any claim of quantitative analysis or gene function performance should not be done with this study. It's an observational study and should be kept that way. Any claim/prediction that is quantitative/transcriptome related throughout the manuscript has to be rewritten
· Line 579 – 607: Experiment and claims are not as per the taxonomic rules (as mentioned above) hence has to be redone and resubmitted.
Overall Comments:
Reviewer acknowledges the genuine effort put by the authors but the current manuscript has too many scientific loopholes to be accepted in its current form as mentioned in the “Major improvements” hence stand rejected.
Author Response
Following is our response to the reviewer’s comments. Shown in black color are the reviewer's comments and the author’s responses appear in red color.
Reviewer # 1:
After long the reviewer came across the manuscript with bench work data backed up by the bioinformatics data and vice versa. This effort to bring the two world together is appreciated
Author's Response: We greatly appreciate your time in reviewing our manuscript thoroughly which facilitated significant improvement; we hope that this iteration is acceptable for publication.
Title of the manuscript is bit misleading and has to be changed, culture-omics is taking the lead here, and then the body of the experiment is framed: the author should address this in the title
Author's Response: Title has been rephrased as suggested.
Line 67 – 88: Author can write this para in a balanced manner explaining the comments at family or genus level phyla is a bit too vague.
Author's Response: We have revised this section to be more coherent.
Line 199 – 215: It's always good to have a simple diagram while explaining material method less know to the public or write it in a language which is conceivable in one read. Reviewer suggest the authors do either of the two
Author's Response: We have revised this section to be more coherent.
Line 216: Microbiome analysis: If Author has used two different set of program one for total metagenome (Author's Response:) and other which is used for amplicon analysis (Qiime) review expect the authors to explain the transition and valid bash/programming codes should be provided with the manuscript.
Author's Response: The bioinformatic pipelines and their workflows are well known and established computational tools. We provide citations to their original works describing their development, including valid bash/programming so we wish not to include this information, which will be redundant.
Line 252: ……..on 16S or 18S rDNA sequencing analysis. :
The statement is very misleading; please use the scientific language to justify your case, be specific of what was used for the sequencing analysis.
Author's Response: The bacterial DNA was sequenced using 16S and the fungal DNA was sequenced using 18S. For coherence, we have have rephrased the sentence to “Bacterial and fungal strains were identified based on 16S and 18S rDNA sequencing analysis, respectively’.
Line 297: What was the concatenation strategy please provide the relevant codes or reference.
Author's Response:
All and every reference of MT2 being novel has not been stated as per the rules laid by the taxonomic society. Please refer to IJSEM fungal novel species reference to execute the same. Using of ANI and dDDH as a speciation marker is only valid for Prokaryotes not for Eukaryotes. Eukaryote taxonomy does not follow those standards, for example, Human, Baboon, and Chimps share more than 98.6% of genome yet they are different species. Reviewer request authors to correct this throughout the manuscript and validate MT2 as new species as per the defined rules.
Author's Response: We concur with the reviewer that the pipeline and approaches we used to evaluate the novelty of strain MT2 are meant for prokaryotes. However, note that we have used the word “potential” to state that most likely the strain is novel, if the same criteria that is sued for bacteria is applied to fungi. However, note that several lines of evidences were used to predict novelty of MT2.
- Firstly, a phylogenetic tree based on manually queried housekeeping genes Tsr1, Cct8, RPB1, and RPB2A, from the genome sequence of strain MT (as shown in figure 8A). A similar approach was recently used to characterize a novel Penicillium spp. isolated from an extremely metal-rich mining site in Russia [Glukhova, L.B.; Frank, Y.A.; Danilova, E.V.; Avakyan, M.R.; Banks, D.; Tuovinen, O.H.; Karnachuk, O.V. Isolation, Characterization, and Metal Response of Novel, Acid-Tolerant Penicillium spp. from Extremely Metal-Rich Waters at a Mining Site in Transbaikal (Siberia, Russia). Microb. Ecol. 2018, 76, 911–924.]. This analysis revealed closest affiliation of strain MT2 with Penicillium janthinellum strain NCIMI366.
- Secondly, further iteration of the evolutionary relatedness of MT2, genome-wide comparisons were run with a cohort of 28 other Penicillium genomes, which confirmed that the closest taxonomic affiliation of MT2 was with P. janthinellum, followed by P. decumbens strain IBT, P. oxalicum and P. subrubescens, respectively (Fig. 8B).
Based on these combined analyses, it appears that MT2 is potentially a new species related to P. janthinellum.
Line 475 – 525: Reviewer would like to stress here that the author have conducted a meta-genomic analysis and any claim of quantitative analysis or gene function performance should not be done with this study. It's an observational study and should be kept that way. Any claim/prediction that is quantitative/transcriptome related throughout the manuscript has to be rewritten
Author's Response: Please note that we performed and presented data using both- amplicon based metagenomics and shotgun metagenomics. The latter was done to infer gene functions, which is a well-accepted technique in the field of microbial ecology. Furthermore, we have not presented data on quantification of gene copy numbers anywhere in the manuscript but have emphasized the relative abundances of taxonomic diversity, which is also well-accepted in the field.
Line 579 – 607: Experiment and claims are not as per the taxonomic rules (as mentioned above) hence has to be redone and resubmitted.
Author's Response: Please note that several lines of evidences were used to predict taxonomy and novelty of MT2 as stated above. We wish to reiterate that our claim is not that the strain is novel but “potentially” novel, which is suggestive and not conclusive.
Moreover, a phylogenetic tree based on manually queried housekeeping genes Tsr1, Cct8, RPB1, and RPB2A, from the genome sequence of strain MT (as shown in figure 8A). A similar approach was recently used to characterize a novel Penicillium spp. isolated from an extremely metal-rich mining site in Russia [Glukhova, L.B.; Frank, Y.A.; Danilova, E.V.; Avakyan, M.R.; Banks, D.; Tuovinen, O.H.; Karnachuk, O.V. Isolation, Characterization, and Metal Response of Novel, Acid-Tolerant Penicillium spp. from Extremely Metal-Rich Waters at a Mining Site in Transbaikal (Siberia, Russia). Microb. Ecol. 2018, 76, 911–924.]. This analysis revealed closest affiliation of strain MT2 with Penicillium janthinellum strain NCIMI366.
Overall Comments:
Reviewer acknowledges the genuine effort put by the authors but the current manuscript has too many scientific loopholes to be accepted in its current form as mentioned in the “Major improvements” hence stand rejected.
Author's Response: We have addressed all your major comments/suggestions and hope that this iteration will be acceptable for publication.

Reviewer 2 Report
In this study Jaswal and co-workers investigated the microbiota associated with uranium-contaminated soil using enrichment cultures and metagenomics. They demonstrated that diffusion chambers and microbial traps were successful in amplifying uranium-tolerant populations, and isolates of key species such as Burkholderia and Penicillium were obtained. Five isolates were demonstrated to deplete uranium from culture medium, indicating bioremediation potential. The metagenomic data confirmed the dominance of these taxa in the original soil and enrichment cultures, and identified potential functional genes involved in uranium-resistance. A whole genome sequence was obtained for one of the Penicillium isolates (strain MT2) and genome comparison suggests this is a novel strain related to P. janthinellum that will be of value for further study.
Comments for Authors
This study is generally well presented - please edit carefully to reduce repetition in the text, be careful not to over-state the significance of clustering dendrograms/PCA results, and improve the legibility of “rainbow” figures.
One thing - do you want to emphasize “High throughput Approach” by having this in the title? This is not really a focus of the paper, so you may be better served with “Metagenomics‐guided Analysis, Isolation and Characterization of Uranium Resistant Microbiota” or “Characterization of Uranium Resistant Penicillium and Burkholderia spp. isolated through Metagenomics‐guided Analysis and Enrichment of Contaminated Soil”.
Specific details:
Line 52: replace “several new approaches have recently been developed,” with “common approaches include”
Line 99: remove “continuously”
Line 121: define abbreviation “SRS” at first use
Line 144: remove “with at least one suggested to be a new fungal genera within the Penicillium group” as you should be stating your aim here, not reporting results.
Please proof read and consolidate your introduction - e.g. Lines 104-106 is a repetition of Lines 80-82, the section from Line 150 is repeating line 93 - 94; line 155 - 156 is repeating line 141 - 142.
Delete the section from line 154 “In the current study…” to line 168 - this is Conclusion and does not belong in Introduction.
Methods section 2.2 - first paragraph (lines 182-192) should be removed as all details are given in greater detail in second paragraph.
Line 285 - why mention assembling PacBio reads when details are only provided for the Illumina sequencing? Provide details if you also did PacBio sequencing.
Lines 318-327 describing the details of GGDC are not necessary.
Line 391 - if the communities are statistically different please report the p-value/ R-value or other supporting evidence. (The dendrogram shows clustering only).
Line 416ff - In contrast to what you have stated about the initial soil samples being similar, the beta diversity plot shows that your soil samples were very different to each other initially, and that the communities converged during the generations of DC/MT culture.
Line 449 - again, show statistical evidence if you want to claim the communities have a statistically significant difference
Line 465 - correct the typo “40 - 40%”
Line 517 - I agree that there is a difference in gene function between the initial soil communities and Generation1+2 of cultivation, and I also agree that there is not much difference between communities obtained from DC or MT method. However you have not demonstrated these differences are “driven by incubation times” - your conclusion in line 522-523 that it is the artificial environment that is responsible is far more likely. Also, be careful with descriptions and conclusions as the PCA analysis (Fig SI- 5B) shows DC Gen1 and DC Gen2 relatively close to each other but MT Gen1 and MT Gen2 are far apart.
Line 567 - Note that the growth profile of MT2 without U does reach a larger diameter than with 8mM or 10mM U, so the growth profile is not the same.
Line 580 - remove the clause “with an N50 contig length of 179,326 bases” from this sentence. Did you close the genome? Please report the N50 contig length and the total number of contigs in the genome in a new sentence.
Line 581 - check genome size as the commas appear to be in incorrect places “341,231,10 bp”
Fig 1A is ok as each colour only appears once per column, but the rainbow colour scheme used here and throughout the paper is confusing. For example, for Fig 1B, only 20 genera are listed in the legend but there are more than 200 coloured bands in the DC Soil column - what are the other ~180 genera? Do not reuse a colour if it is a different taxon. To show a long tail of low-abundance genera a plot of relative abundance (y-axis) vs genera (x-axis) or a table would be better.
Fig 3 - the colours make the figure impossible to interpret. Make a table instead.
Fig 6 legend - is the data in panel A at 1000uM or 2000uM U ?
Fig 8 - Tree (A) is informative and its construction is well described in the methods. Tree (B)’s construction is not clearly described - also it is not rooted, has no bootstraps reported, and does not add much additional information. Please remove Tree (B).
Fig SI-6A and 6B - please re-do each tree to include an outgroup species and use this to root the tree. Do not report bootstraps below 75% confidence.
Fig S1-7 - Please use Mauve (or similar program) to order MT2’s contigs to align with the reference before creating the plot as this should highlight regions of similarity/difference with the reference.
Author Response
Following is our response to the reviewer’s comments. Shown in black color are the reviewer's comments and the author’s responses appear in red color.
Reviewer # 2:
This study is generally well presented - please edit carefully to reduce repetition in the text, be careful not to over-state the significance of clustering dendrograms/PCA results, and improve the legibility of “rainbow” figures.
Author's Response: We greatly appreciate your time in reviewing the manuscript thoroughly which significantly improved overall presentation. The revised iteration does not include repetitive text, results have been framed in context without overstating them and figure quality have been improved. We hope that this iteration is acceptable for publication.
One thing - do you want to emphasize “High throughput Approach” by having this in the title? This is not really a focus of the paper, so you may be better served with “Metagenomics‐guided Analysis, Isolation and Characterization of Uranium Resistant Microbiota” or “Characterization of Uranium Resistant Penicillium and Burkholderia spp. isolated through Metagenomics‐guided Analysis and Enrichment of Contaminated Soil”.
Author's Response: We have revised the title to avoid high-throughput claim.
Line 52: replace “several new approaches have recently been developed,” with “common approaches include”
Author's Response: The authors have modified the sentence as the reviewer suggested.
Line 99: remove “continuously”
Author's Response: Note that there’s continuous transport of nutrients through the bottom membrane of DC/MT. However, the authors have deleted the word ‘continuously’ as suggested.
Line 121: define abbreviation “SRS” at first use
Author's Response: SRS has been defined in this sentence.
Line 144: remove “with at least one suggested to be a new fungal genera within the Penicilliumgroup” as you should be stating your aim here, not reporting results.
Author's Response: This section has been revised.
Please proof read and consolidate your introduction - e.g. Lines 104-106 is a repetition of Lines 80-82, the section from Line 150 is repeating line 93 - 94; line 155 - 156 is repeating line 141 - 142.
Author's Response: We have now carefully combed through the manuscript to avoid repetition and redundancy.
Delete the section from line 154 “In the current study…” to line 168 - this is Conclusion and does not belong in Introduction.
Author's Response: Writing styles can vary depending on the nature of publication. We put our work in context to the little information that exists on metagenomics-guided isolation of microorganisms including the use of diffusion chambers and in summation of introduction, it made more sense to include some of our conclusions.
Methods section 2.2 - first paragraph (lines 182-192) should be removed as all details are given in greater detail in second paragraph.
Author's Response: This section has been modified per your recommendation.
Line 285 - why mention assembling PacBio reads when details are only provided for the Illumina sequencing? Provide details if you also did PacBio sequencing.
Author's Response: PacBio sequencing was added as a mistake as the sequencing was performed using Illumina only. We have removed PacBio from the manuscript.
Lines 318-327 describing the details of GGDC are not necessary.
Author's Response: Details on GGDC have been deleted.
Line 391 - if the communities are statistically different please report the p-value/ R-value or other supporting evidence. (The dendrogram shows clustering only).
Author's Response: The word ‘statistically different’ has been replaced by ‘distinctly different’, as dendrogram clearly shows the differences between the samples being discussed.
Line 416ff - In contrast to what you have stated about the initial soil samples being similar, the beta diversity plot shows that your soil samples were very different to each other initially, and that the communities converged during the generations of DC/MT culture.
Author's Response: We agree and have revised this section according to your observation.
Line 449 - again, show statistical evidence if you want to claim the communities have a statistically significant difference
Author's Response: The word ‘statistically different’ has been replaced by ‘distinctly different’, as dendrogram clearly shows the differences between the samples being discussed.
Line 465 - correct the typo “40 - 40%”
Author's Response: This was a typographical error and has been corrected to “40-48%”.
Line 517 - I agree that there is a difference in gene function between the initial soil communities and Generation1+2 of cultivation, and I also agree that there is not much difference between communities obtained from DC or MT method. However you have not demonstrated these differences are “driven by incubation times” - your conclusion in line 522-523 that it is the artificial environment that is responsible is far more likely. Also, be careful with descriptions and conclusions as the PCA analysis (Fig SI- 5B) shows DC Gen1 and DC Gen2 relatively close to each other but MT Gen1 and MT Gen2 are far apart.
Author's Response: We agree that the incubation times may not be as significant as the growth conditions presented by the DC/MT conditions and have revised the sentence accordingly. We have also revised the section on PCA analysis, as per your observation and suggestions.
Line 567 - Note that the growth profile of MT2 without U does reach a larger diameter than with 8mM or 10mM U, so the growth profile is not the same.
Author's Response: The authors agree with the reviewer’s observation and the discussion of the said result has been modified accordingly.
Line 580 - remove the clause “with an N50 contig length of 179,326 bases” from this sentence. Did you close the genome? Please report the N50 contig length and the total number of contigs in the genome in a new sentence.
Author's Response: This sentence has been revised per your suggestion. The genome sequence was not closed as this was not the main focus of this part of the study.
Line 581 - check genome size as the commas appear to be in incorrect places “341,231,10 bp”
Author's Response: Thank you for bringing this to our notice; it has been rectified.
Fig 1A is ok as each colour only appears once per column, but the rainbow colour scheme used here and throughout the paper is confusing. For example, for Fig 1B, only 20 genera are listed in the legend but there are more than 200 coloured bands in the DC Soil column - what are the other ~180 genera? Do not reuse a colour if it is a different taxon. To show a long tail of low-abundance genera a plot of relative abundance (y-axis) vs genera (x-axis) or a table would be better.
Author's Response: As the reviewer noted, there are more than 200 genera identified in the soils as shown in figure 1A, and hence there is repetition of color. We do not wish to show this data in a table format since the visual depiction makes for a better presentation. Moreover, the main point of predominance of proteobacterial phyla along with Burkholderia/ Rhodanobacter genera are clearly shown in the figure so we wish to retain this figure without any changes.
Fig 3 - the colours make the figure impossible to interpret. Make a table instead.
Author's Response: Similar to bacterial communities, for the fungal phyla and genera, figure clearly shows dominance of Ascomycota phyla and Penicillium genera, hence we do not wish to show this data in a table format.
Fig 6 legend - is the data in panel A at 1000uM or 2000uM U ?
Author's Response: The experiment was performed at both 1000µM and 2000µM. The data shown in Fig 6A is only from 2000µM. The authors have modified the sentence to make it coherent.
Fig 8 - Tree (A) is informative and its construction is well described in the methods. Tree (B)’s construction is not clearly described - also it is not rooted, has no bootstraps reported, and does not add much additional information. Please remove Tree (B).
Author's Response: As suggested by the reviewer, Fig 8B has been removed from the mansucript.
Fig SI-6A and 6B - please re-do each tree to include an outgroup species and use this to root the tree. Do not report bootstraps below 75% confidence.
Author's Response: The authors have modified the figures according to the reviewer’s suggestion.
Fig S1-7 - Please use Mauve (or similar program) to order MT2’s contigs to align with the reference before creating the plot as this should highlight regions of similarity/difference with the reference.
Author's Response: We have used blastn to run comparisons between the MT2 genome and other genomes so MAUVE is not necessary for this data representation.

Reviewer 3 Report
The authors present their scientifically-relevant and appropriately planned research in a well-written and structured article. In it, they assess the ability of two modern isolation methods (diffusion chamber and microbial trap) to retrieve ecologically relevant microorganisms, comparing the results of their use with the original soil microbiota.
Here I include some suggestion and comments, as well as minor corrections to the manuscript:
- Taxonomically curated databases EzTaxon or EZ-Taxon-e would be better suited than NCBI Blast database for the evaluation of Taxonomic assignation of PCR-derived 16S rRNA sequences.
- Accession numbers starting by SUB (in 2.10 and 2.11) give no result in any of NCBI databases.
- Line 387: Uranium
- Line 395: This is not a comment/discussion on what the first two sentences indicate.
- Line 424: Although
- Line 453: “…time not as much as…”: Is this an erratum or do authors indicate that, contrary to other cases stated in the manuscript, here time is less important than Isolation technique (DC, MT)
- Line 465: Number 40 seems to be duplicated
- Line 509: Delete comma before the stop.
- Line 538: Exactly how much time is “prolonged incubations”?
- Line 615-616: It could be interesting to include the fact that the microbiota of uraniferous anaerobic ecosystems have already been characterised, but aerobic ones have been mostly disregarded until now.
- Line 619: “compared” is duplicated
- According to authors, what could be the reason for the high alpha diversity in generation 2 (Gen 2), as compared to Gen 1?
- To reduce the number of figures in the article, I would suggest to merge figures 1 and 3, and move figures 7 to 9 to supplementary material
- Figure 8: Authors may consider highlighting the here-isolated strain in color/bold.
Author Response
Following is our response to the reviewer’s comments. Shown in black color are the reviewer's comments and the author’s responses appear in red color.
Reviewer # 3:
Here I include some suggestion and comments, as well as minor corrections to the manuscript:
Taxonomically curated databases EzTaxon or EZ-Taxon-e would be better suited than NCBI Blast database for the evaluation of Taxonomic assignation of PCR-derived 16S rRNA sequences.
Author's Response: Thanks for bringing these pipelines to our attention but NCBI is the gold standard used for several decades. There’s no reason to redo the taxonomy.
Accession numbers starting by SUB (in 2.10 and 2.11) give no result in any of NCBI databases.
Author's Response: The accession number has been revised to PZKB00000000 (https://www.ncbi.nlm.nih.gov/nuccore/PZKB00000000).
Line 387: Uranium
Author's Response: Uranium has been abbreviated as U and used interchangeably as U or uranium throughout the manuscript, which is widely accepted.
Line 395: This is not a comment/discussion on what the first two sentences indicate.
Author's Response: This section has been revised.
Line 424: Although
Author's Response: The authors have corrected this typographical error.
Line 453: “…time not as much as…”: Is this an erratum or do authors indicate that, contrary to other cases stated in the manuscript, here time is less important than Isolation technique (DC, MT)
Author's Response: For both bacterial and archaeal shifts shown in our study, it appears that incubation times (a.k.a generations) influenced growth relative to the membrane size difference between DC-MT. This is a consistent trend and is stated in the manuscript on pages 17 and 19.
Line 465: Number 40 seems to be duplicated
Author's Response: The authors have corrected this typographical error.
Line 509: Delete comma before the stop.
The authors have corrected this typographical error.
Line 538: Exactly how much time is “prolonged incubations”?
Author's Response: Authors have changed the sentence to include the time period of the incubations.
Line 615-616: It could be interesting to include the fact that the microbiota of uraniferous anaerobic ecosystems have already been characterised, but aerobic ones have been mostly disregarded until now.
Author's Response: We agree; this was stated in the previous and current versions in the closing paragraph of the study: “In summation, the ability of bacterial strains to survive in uraniferous ecosystems has been well demonstrated for anaerobic ecosystems, but the potential of aerobic microorganisms to biomineralize uranium has been largely ignored and hence remains understudied”.
Line 619: “compared” is duplicated
Author's Response: The authors have corrected this typographical error.
According to authors, what could be the reason for the high alpha diversity in generation 2 (Gen 2), as compared to Gen 1?
Author's Response: As we state in the revised manuscript: This can also reflect the initial stress posed by the artificial growth conditions being presented by DC/MT. However, the alpha diversity almost reached back up to the level of soils in both DC/MT Gen 2, suggesting that the soil microbiota is potentially resilient and overcame the initial stress of the artificial growth conditions, as was observed by reduction of alpha diversity in Gen1 experiments.
To reduce the number of figures in the article, I would suggest to merge figures 1 and 3, and move figures 7 to 9 to supplementary material
Author's Response: Figures 1 and 3 show bacterial and fungal diversity at the phylum and genera levels and hence contain different information which cannot be merged into a single figure. Furthermore, figures 7 and 9 depict important genomic features of the potentially novel penicillium strain isolated and characterized in this study. Hence, we wish to retain these figures in the main document.
Figure 8: Authors may consider highlighting the here-isolated strain in color/bold.
Author's Response: Authors have bolded the isolated strain, so as to make it easily visible among the other strains.
